# Exploring 3D miniatures with action simulations by finger gestures: Study of a new embodied design for blind and sighted children

**Dannyelle Valente** [1,2][☉] *, **Amaya Palama** [1][☉], **Edouard Gentaz** [1][☉]

**1** Faculty of Psychology and Educational Sciences, University of Geneva, Geneva, Switzerland,
**2** Department of Developmental Psychology, University of Lumière Lyon 2, Bron, France

☉ These authors contributed equally to this work.
* dannyelle.valente@univ-lyon2.fr

**Data Availability Statement:** All relevant data are within the manuscript and its Supporting Information files.

## Abstract

Tactile books for blind children generally contain tactile illustrations referring to a visual world that can be difficult to understand. This study investigates an innovative way to present content to be explored by touch. Following embodied approaches and evidence about the advantages of manipulations in tactile processing, we examined 3D miniatures that children explored using their middle and index fingers to simulate leg movements. This "Action simulations by finger gestures–ASFG" procedure has a symbolic relevance in the context of blindness. The aim of the present study was to show how the ASFG procedure facilitates the identification of objects by blind and sighted children. Experiment 1 examined the identification of 3D miniatures of action objects (e.g. the toboggan, trampoline) by 8 early blind and 15 sighted children, aged 7 to 12, who explored with the ASFG procedure. Results revealed that objects were very well identified by the two groups of children. Results confirmed hypotheses that ASFG procedures are relevant in the identification process regardless of the visual status of subjects. Experiment (control) 2 studied identification of tactile pictures of same action objects by 8 different early blind and 15 sighted children, aged 7 to 12. Results confirmed that almost all objects obtained lower recognition scores in tactile pictures than in 3D miniatures by both groups and showed surprisingly higher scores in blind children than in sighted children. Taken together, our study provides evidence of the contribution of sensorimotor simulation in the identification of objects by touch and brings innovative solutions in book design for blind people. Moreover, it means that only the ASFG procedure has a very inclusive potential to be relevant for a larger number of subjects, regardless of their visual skills.

## Introduction

Tactile illustrated books for blind children generally contain tactile illustrations that are transferred from visual illustrations through different techniques such as thermoforming or

**Funding:** Grant: Edouard Gentaz (EG) Swiss National Science Foundation (FNS) FNS N° 100019_162688 http://www.snf.ch/en/Pages/default.aspx The funders had no role in study design, data collection and analysis, decision to publish, or preparation of the manuscript.

**Competing interests:** The authors have declared that no competing interests exist.

textures. Several studies have shown that this content based on visual reality can be difficult to understand for children and adults with visual impairments [1–8].

Comparing vision and touch, several researchers now support the idea of a functional specialization of the senses, instead of a hierarchical vision of the senses [9–12]. Each sense excels in processing certain properties. Thus, the haptic sense is very efficient in the perception of the texture and hardness of materials, but it is less so regarding spatial qualities. This specialization is probably due to the simplicity of optimal exploratory procedures for perceiving texture or hardness, while those adapted to geometric properties require movements that are coordinated in time and space [9].

In studies testing blind adults' identification of 2D representations of familiar objects (utensils, animals, etc.), the success rate was less than 20% [13–17]. This rate can be increased to 40%, however, if researchers provide information about the category of the object (whether it is a fruit or an animal, for example) [1,17].

Several authors explain this difficulty by pointing out these subjects' limited ability to generate visuospatial imagery [14,16,18] and/or their lack of familiarity with the visual conventions of drawing [13,15,16,19,20]. Thus, Lederman et al. [14] have highlighted the role of a process called 'visual mediation' for the identification of tactile drawings. When the subject touches an image, haptic sensors must be mentally translated into a visual image on which its identification depends. For authors, it explains why blindfolded sighted adults performed better than congenitally blind subjects on tactile identification tests of tactile drawings of everyday objects [14,16]. The role of visuospatial imagery has also been highlighted in the processing of non-figurative two-dimensional patterns by blindfolded sighted subjects [18].

Therefore, studies have also shown that performances were influenced by expertise with bimanual exploratory processes [21]. Thus, late blind children performed better than their early blind and blindfolded sighted counterparts due to their previous visual experience associated with a specialization in the bimanual exploratory process because of their blindness experience [13,16]. Theurel et al. [1] also showed an effect of practice in early blind children's performance. Children who regularly or moderately use tactile illustrations at school or at home performed globally better in the identification task than participants who did not practice with tactile illustrations or did so infrequently.

Other studies have shown that the illustration technique might also influence identification rates. Theurel et al. [1] showed that blind children identify an object more easily through textured illustrations than through raised lines and thermoformed illustrations. The texture technique involves handcrafting a collage of several textures (paper, fabrics, etc.). The texture and thermoforming technique were recognized more often than tactile line drawings, in which surfaces of objects were simply bound [15]. Textures have an advantage over thermoform because they might include elements that are similar to real material properties of objects (fur to represent the hair of an animal, foam to represent a banana peel, etc. [1]).

However, a current problem in illustrations produced by assembling textures is that these materials are more often used to "colour" different formal areas of the visual images rather than to simulate real textures of objects. For instance, designers might use textures to differentiate the black stripes of a zebra but with materials that do not refer to the texture of an animal at all. Despite the use of several textures, some content remains anchored in visually realistic conventions [22]. This may explain why blind subjects' tactile identification rates of illustrations with textures are still low (35.87% in the Theurel et al. [1], experiment), although better than with other techniques.

To attenuate difficulties in tactile picture identification by blind children, Thompson, Chronicle and Collins [16] have proposed a novel design, called 'Texyform', which applies principles of 3-D object recognition to 2-D tactile picture design. Three variations of the same

texture were used to represent vertical and horizontal picture elements as well as indicate when surfaces were rounded or cylindrical in some way, as opposed to flat. Results showed that early blind participants improved their identification from 12.5% with visually realistic pictures to 50% with TexyForm pictures. In the same way, Bara and his colleagues [23,24] showed that 3D elements or manipulations improve story processing by blind children in tactile books. In a study analyzing a book-reading activity using illustrations with textures and "3D illustrations" (use of miniature objects to illustrate the story), Bara [23] showed that children used a wider variety of exploratory procedures with "3D illustrations" than illustrations with textures, which suggested that they were able to collect more information with the "3D illustration" technique. Results obtained in a more recent study showed that when children with visual impairments, specifically congenitally blind children, have the possibility to manipulate miniature objects, it enhances their processing of information in a story [24].

The present study investigates a new way and a more embodied design to represent objects in tactile books. The idea is to use 3D miniatures that children explore using two fingers to simulate leg movements in interaction with real objects. Two fingers/legs perform various actions upon miniature 3D scenes (jumping on a trampoline, climbing stairs, etc.). Following evidence about the advantages of manipulations and tactile processing of 3D elements in tactile books [16,24] this novel design adds a motor component to the exploration by gestures. This new tactile exploration procedure is called "Action Simulations by Finger Gestures–ASFG" [25].

Our experience of the environment is part of an exploration activity where perception and motor skills are strongly involved. Embodied approaches to cognition [26–28] argue that our sensorimotor experiences with real objects contribute to perceptual and conceptual processes. For example, the concept *stairs*, rather than being an abstract or arbitrary representation of its components, is made up of the simulation of our real experience of going up and down stairs [26]. In the same way, several studies in neuroscience and psychology have first shown that an imagined action reproduces parameters very close to those of the perceived action [29]. Studies in neuroscience have shown that brain areas of the motor cortex are activated when subjects are asked to read action verbs such as "run" [30]. Other studies have shown that the reactivation of motor components involved in interactions with objects (enactment effect) can facilitate the learning and memorization of concepts [for a review see 31,32]. Research in the field of language, reading and mathematics has also shown that real actions performed on objects and/or making gestures to mimic a concept such as "opening a door" helps memorization and comprehension of content [33–36].

Furthermore, studies on sensory integration [37,38] have shown that lack of vision can alter the intermodal calibration necessary for spatial discrimination in blind children [for a review see 39]. Thus, compensation via other intact senses associating for example auditory and motor information can help blind children to process spatial information [38]. Gappagli and colleagues have tested a technological device (ABBI—Audio Bracelet for Blind Interaction), which emits a sound when a movement occurs, with 3–7 year-old children with congenital blindness or low vision. After 3 months of training, authors found that the spatial performance of congenitally blind but not low vision children is improved. Results indicated that in the absence of sight, the association of sensorimotor data with other intact senses is of great importance.

In this perspective, we hypothesized that engaging the body in perceptual processes would also be an approach with great potential for designing educational intervention materials for blind children. This is because we are working on simulations of sensorimotor experiences that are also accessible to these children. Giving them the opportunity to simulate real experiences with objects (opening the front door, climbing the stairs) in the book instead of touching

representations of the formal appearance of an object seen from a distance (the facade of a house, the profile silhouette of stairs) could facilitate identification of illustrations by blind readers [22].

Thus, in a recent study, we [25] examined whether the simulations of actions performed on objects using gestures (Action simulations by fingers gestures -ASFG) held the same symbolic meaning to blind and sighted individuals. We also evaluated if patterns of ASFG are the same for blind and sighted people. For that, we asked blindfolded sighted adults, early blind adults and late blind adults to produce ASFGs of 18 action concepts (i.e. slide on a toboggan). ASFGs produced by three groups of encoders were videotaped and analyzed by sighted decoders in a visual recognition task. Results showed that gestures produced by the three groups were very well recognized by sighted decoders. The recognition rate by type of action was also very similar across groups. Indeed, the same motor pattern is found in the sighted and blind adults' simulations. Few differences were found referring to illustrative components of ASFG (i.e., how the gesture is seen by the interlocutor) unknown to those who were blind from birth. Thus, some of their simulations were less recognized by sighted people and the late blind. In spite of these differences in terms of the appearance of the gesture, the results have shown that the procedure for simulating a real action by finger gestures has symbolic relevance for people with and without visual experience.

Based on these results, 7 action objects engaging ASFG were selected for the present study. Our aim at this point was to examine the identification of 3D action objects (like the toboggan or the swing) by blind and sighted children by exploring them through the procedure of ASFG. We also examined the contribution that this simulation procedure could make in relation to a more common exploration procedure of 2D tactile pictures. Two experiments were conducted. In Experiment 1 (experimental), 8 early blind children and 15 sighted children, aged 7 to 12 were asked to identify 3D miniatures of 7 action objects by exploring them with an ASFG procedure. We hypothesized that the ASFG procedure should facilitate the identification of objects, whatever the visual status of children. In Experiment 2 (control), 8 different early blind children and 15 sighted children, aged 7 to 12, were asked to identify the same action objects depicted in 2D by a usual technique of texture. We hypothesized that visuospatial ability [14,16,18] and expertise with visual conventions [1,13,15] enabled by visual experience, should make the identification of textured pictures more difficult.

## Experiment 1: Identification of 7 action objects by early blind and sighted children engaging the ASFG procedure

The aim of this experiment was to study if 3D miniatures engaging ASFGs would be well identified by children by the fact that they supposedly activate an embodied component. In addition, we examined if the performances in identification vary according to participants' visual status. Because of similarities in ASFG production found in blind, visually impaired and sighted adults [25], no difference was expected between sighted and blind children in this identification task.

## Method

### Participants

A total of 23 children participated in Experiment 1: 8 early blind children and 15 blindfolded sighted children. The group of early blind children was composed by 4 girls and 4 boys (M = 8.1 years, SD = 2.2, range 6 to 12 years) and the group of sighted children by 8 girls and 7 boys (M = 7.0 years, SD = 0.7, range 6 to 8 years).

**Table 1. Characteristics of early blind children who participated in Experiment 1: Identification of action objects engaging the ASFG procedure.**

| Code | Gender | Category (ICD-10, 2016) | Age (in years) | Cause of deficit |
|---|---|---|---|---|
| 1 | M | 5 | 7 | Norrie disease |
| 2 | M | 5 | 7 | Leber's amaurosis |
| 3 | M | 4 | 10 | Retinopathy |
| 4 | F | 5 | 6 | Microphthalmia |
| 5 | F | 4 | 6 | Microphthalmia |
| 6 | F | 5 | 12 | Retinopathy |
| 7 | M | 4 | 10 | Alstrom Syndrome |
| 8 | F | 5 | 7 | Retinal Dysplasia |

Table 1 presents characteristics of early blind children who participated in Experiment 1. In the group of early blind children, all participants were congenitally blind or had been blind since birth or the very first years of life. They are also included in one of the three categories of Blindness (Categories 3, 4 or 5) of the International Statistical Classification of Diseases and Related Health Problems (ICD-10 revision, WHO 2016). Children were recruited in 6 special educational centers in French-speaking Switzerland and France. They attended regular classes or *ULIS classes* (i.e., Special classes for pupils with disabilities in regular school in France) and were regularly supported by specialist teachers who provided intervention in the classroom or a special educational center. The group of sighted children was recruited in an Elementary School in Geneva. The sighted children had normal or corrected to normal vision and were typically developing.

Due to the specificities of developmental trajectories and differences in the care of children with visual impairments, early blind children were matched with the control group of sighted children by scholar age and not by chronological age. Both groups attended the 1st or 2nd grade of elementary school. Parents or guardians provided signed written informed consent forms for their children to participate in the experiment. This study respects ethical principles for research involving human subjects (World Medical Association Declaration of Helsinki) and was approved by the Swiss Ethics Committee (project number 2015–00183 (15–297).

## Stimuli

3D miniatures of the 7 action objects and a familiarization board were designed for Experiment 1 (see Fig 1). Items to test were selected on the basis of the results of a previous study already mentioned examining ASFGs produced by blind and sighted adults [25]. The choice was made on the basis of two criteria: 1) the prototyping criterion (the ASFGs produced by the 3 groups of adults obtained a rate of identification of greater than 50% by the sighted judges) and 2) a tangibility criterion (some ASFGs are trajectories to be carried out without a tangible object, i.e. Jumping on one leg or Walking backwards). The ASFG selected needed to be reproducible in 3D miniature and also be represented as 2D content. Thus, the action objects included are the Swing, Bicycle, Toboggan, Trampoline, Merry-go-round, Roller Skate and Stairs.

In order to determine the children's frequency of contact with the action objects included in the test, we asked the adult responsible for each child to complete a 5-point scale (0—*Never* to 4—*Everyday*) questionnaire. The mean global level of frequency of contact is 2.18 (SD = 1.08) for sighted children and 1.71 (SD = 1.15) for blind children. The order of frequency by action object is almost the same for both groups of participants (see S1 Appendix in S1 Table). Both groups engage more frequently with the Stairs, Slide, Swing and Bike. Roller skating was the activity that both groups do the least. In order to determine the effect of

## Familiarization board

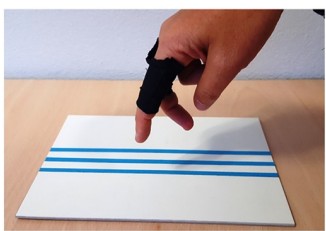
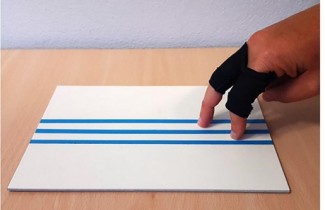

| **3D miniatures of action objects** | **ASFGs made with children during the guidance phase (if not identification)** |
|---|---|

Swing

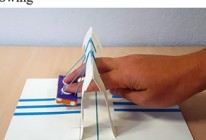
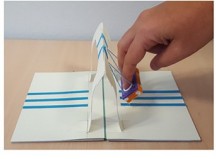

Place both fingers on the swing bench and swing (2x)

Bicycle

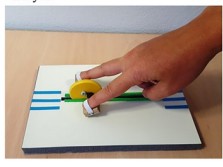
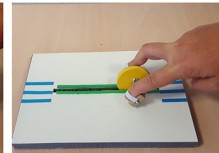

Move forward on the rail with one finger on each pedal, making interspersed circular movements (2X)

Toboggan

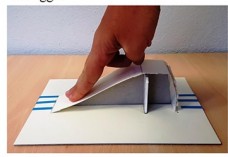
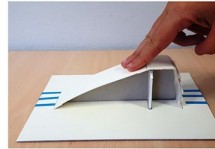

Climb the small ladder with both fingers and slide down (2x)

Trampoline

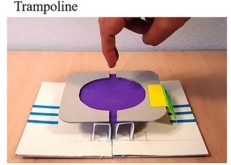
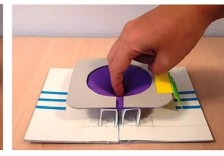

Climb the small ladder with both fingers and jump on the elastic surface (2x)

Merry-go-round

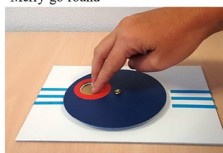
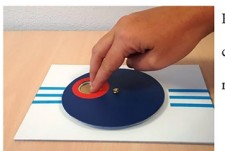

Place both fingers in the small circle and turn the merry-go-round (2x)

Roller skate

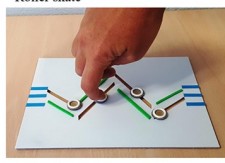
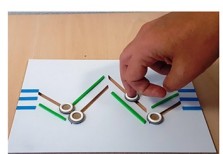

Place the middle finger in the small circle, advance to the left, then do the same with the index finger (2x)

Stairs

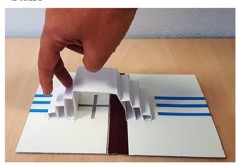
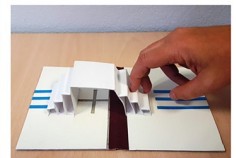

Go up the three steps, cross the platform, and go down the three steps (2x)

**Fig 1. Stimuli included in Experiment 1.** Fig 1A. Familiarization board, Fig 1B. Swing and Bicycle, Fig 1C. Toboggan and Trampoline, Fig 1D. Merry-go-round and Roller Skate and Fig 1F. Stairs. ASFGs made by children during the guidance phase were described (right) for each 3D miniature (see Procedure).

frequency of contact in identification scores, we have established a between subjects factor taking as a basis the global average frequency per action object for each group (blind and sighted). Children below the average for their group had *No or Infrequent* contact whereas children above the average had *Regular or Moderate* contact.

Stimuli were produced by the non-profit publishing house Les Doigts Qui Rêvent. Each miniature measured 21 cm X 14.5 cm. They were fixed in a board containing three parallel tactile lines (see Fig 1). The children had to follow it with their fingers as they would with their feet when following a real tactile path for blind people. A familiarization board (Fig 1A) was also designed to familiarize children with the ASFG procedure before beginning the identification task. The same tactile path was found on each test board (Fig 1B–1F) in order to direct fingers to the 3D miniatures to be explored.

## Procedure

The identification task was administered individually at school or at an educational center. For sighted children, we used a blackout curtain during the task to restrict identification to the tactile modality.

Before the identification task itself, the experimenter introduced participants to the procedure of action simulation with two fingers. This familiarization phase was important to ensure that the principle of simulating leg actions by fingers was well understood by both groups of participants. In the environment of sighted children, many games and nursery rhymes use two fingers as a substitute for a character, but it is possible that this symbolic play is not familiar to early blind children. Studies about symbolic play in the context of blindness [40] showed that blind children use objects in miniature or gestures less spontaneously than sighted children as substitutes for a real object or real actions.

In order to make sure that all children adhered to the proposed symbolic approach ("my two fingers are the two legs of a character") the experimenter used a familiarization board with the same dimensions as the test boards (see Fig 1A). The familiarization board was empty and contained only a tactile path. At first, the experimenter indicated to the child the two fingers (index and middle fingers) that he should use for the task and gave the instruction: "I am asking you to pretend that these two fingers are the two legs of a character". Depending on the laterality indicated by the child, the task was performed with either the right or the left hand. To reinforce the symbolism "two fingers-two legs" the experimenter asked the child if he or she wanted to give a name to the character and if he or she wanted to dress the two fingers-legs with a small pair of pants. The experimenter then asked the child to show how his character "walks", "runs" and "jumps" on the familiarization board. The task did not begin until the child demonstrated that he/she understood the procedure. Although the simulation procedure was implemented much more spontaneously by sighted children, the blind children also quickly understood the instructions and all participants performed ASFGs of walking, running and jumping with both fingers without difficulty.

In the identification task, participants were told to move along with two fingers through the tactile path that would lead them to "something to do" and the goal was to guess what it was. 3D miniatures of action objects were presented in a random order across participants. A category was given for each action object: "To play in a playground" for the Toboggan, Merry-go-round, Trampoline and swing and "To move" for the Bicycle and Roller Skate. In order to avoid giving the category "To go up" for Stairs which itself defines the object, the experimenter

indicated that this can also be in a park for fun but also in other places such as at school and at home. The children were told to try to give an answer even if they were not sure.

If children were unable to identify the action object after one minute of exploration, the experimenter achieved the expected trajectory once by holding the child's hand. Fig 1 describes the ASFG performed with the child for each object in the guidance phase. No feedback was given, regardless of whether or not the answer was correct. If they were unable to identify an action object even in the guidance phase, participants were told to inform the experimenter when they wished to stop exploring.

We therefore calculated an identification score (max score is 2.00) for each child and for each action object. Two (2) points were awarded if the child gave a correct response without any help, a single (1) point if they gave the correct response after guidance, and no (0) point if no response or an incorrect response was given. An analysis of variance was performed with identification scores. The accepted level of significance for ANOVAs is $p = 0.05$.

## Results

In order to determine the effect of frequency of contact with action objects in identification scores, a one way ANOVA was performed on the identification scores with the factor frequency of contact (0 = never or infrequent and 1 = moderate or frequent). Frequency of contact was not significant for either action object tested (all p. > 0.5). In the early blind group, a significant effect was found only for the Trampoline, p = 0.04, (all p.>0.5 for other action objects). However, it is necessary to note that only 1 of the 8 children had "moderate or frequent" contact with Trampolines. Surprisingly, this child was the only one of the group who did not identify the Trampoline. The small size of our sample does not allow us to draw any conclusive results on the effect of frequency of contact in identification scores. Therefore, the frequency will not be considered as a between-subject factor in our subsequent ANOVAs.

A mixed-design ANOVA was performed on identification scores with action objects as within subjects factors (the Swing, Bicycle, Toboggan, Trampoline, Merry-go-round, Roller Skate and Stairs) and group as between subject factors (early blind vs sighted). Mauchly's test indicated that the assumption of sphericity had been violated, therefore degrees of freedom were corrected using Greenhouse-Geisser estimates of sphericity ($\varepsilon = 0.61$). The main effect of group was not significant ($F(1,21) = .94$, $p = .34$). Sighted children obtained a very good identification score of 1.38 (SD = 0.14) very near to the score of 1.29 (SD = 0.33) obtained by blind children. Therefore, results revealed a main effect of action object on identification scores ($F(1, 29) = 37.2$, $p < .001$) and an interaction between action object and group ($F(1,29) = 4.41$, p = .004). A post-analysis using Bonferroni showed only a significant difference in scores obtained in the two groups for the Trampoline ($p < .001$). Indeed, blind children were globally good at identifying the trampoline (score: 1.25/2.00, SD = 0.71) but not as good as sighted children who obtained the maximum score (2.00/2.00). As presented earlier, the frequency of contact with the trampoline does not seem to play any role here.

Table 2 shows identification scores for each action object by each group. For both groups, the action object which obtained the best score is the Toboggan (all participants obtained maximum scores). Both groups also obtained very good identification scores for the Stairs and Swing. The two groups obtained lower scores on the action objects Bicycle and Roller Skate.

## Discussion

Our main result is that the factor group is not significant in identification scores which would confirm our hypothesis that common sensorimotor experiences are involved in ASFG procedures and that visual experience has less importance in this case.

**Table 2. Identification scores (max 2.00) and (SD) in sighted (N = 15) and early blind children (N = 8) for each action object explored with the ASFG procedure.**

| Action object | Sighted | | Early blind | | Total | |
|---|---|---|---|---|---|---|
| Swing | 1.67 | (0.49) | 1.75 | (0.71) | 1.70 | (0.56) |
| Bicycle | 0.33 | (0.49) | 0.25 | (0.46) | 0.30 | (0.47) |
| Toboggan | 2.00 | (0.00) | 2.00 | (0.00) | 2.00 | (0.00) |
| Trampoline | 2.00 | (0.00) | 1.25 | (0.71) | 1.74 | (0.54) |
| Merry-go-round | 0.93 | (0.80) | 1.63 | (0.74) | 1.17 | (0.83) |
| Roller Skate | 0.73 | (0.46) | 0.50 | (0.54) | 0.65 | (0.49) |
| Stairs | 2.00 | (0.00) | 1.63 | (0.74) | 1.87 | (0.46) |
| Total | 1.38 | (0.14) | 1.29 | (0.33) | | |

Overall, identification scores of action objects with the ASFG procedure were high for blind and sighted children. Scores varied according to each action object. The fact that the Bicycle and Roller skate obtained lower identification scores in the set of miniatures tested seemed mainly linked to the design of the tactile devices. In the exploration of these two devices, children had difficulty understanding where they should place each finger and the experimenter also had to guide them to find the right way to execute the ASFG expected. Thus, the failure here seems to be related to specific ergonomics problems. Both devices require participants to find the correct location or place their fingertips on the book and walk forward while hooked to the device.

Both groups scored well for the other action objects. Only a difference between groups was found for the Trampoline. Indeed, our data show that blind children still scored well in terms of identification (score: 1.25) but less so than sighted children (score: 2.0). Of the 8 blind children, 3 gave the correct answer before the guiding phase, 4 after the guiding phase and 1 participant failed to identify the object, whereas in the sighted group all managed to find the object without going into the guiding phase.

The small size of our sample does not allow us to consider frequency as a between-subjects factor. On the other hand, if global scores of frequencies by action object are taken into account we notice that the action objects obtaining the highest identification scores are those that the children have more frequent contact with (the Toboggan, Stairs and Swing, see S1 Appendix in S1 Table). The little contact with Roller Skates by both groups may also explain the low scores obtained for this action object in addition to the fact that this device was more complex to explore.

## Experiment (control) 2: Identification of textured pictures of 7 action objects by early blind and sighted children

The second experiment involved the same 7 action objects and a different group of 8 early blind children and 15 sighted children, aged 7 to 12. The aim of Experiment (control) 2 was to confirm that identification of tactile pictures of these objects depicted in 2D using the usual technique of textures should be difficult and be modulated by the visual status of the participants. The results of Experiment 1 revealed that these action objects in 3D explored with the ASFG procedure were generally well identified by both groups. Examining identification of textured pictures of the same action objects may provide additional evidence on the effect of sensorimotor simulation on the identification of the items in Experiment 1. According to studies comparing performances of early blind and blindfolded sighted children and adults [14,16,18] a group effect on low scores of identifications was expected in Experiment 2. Sighted children are expected to achieve better results than early blind children due to visual imagery

**Table 3. Characteristics of early blind children and children with low vision who participated in Experiment 2: Identification of action objects in textured pictures.**

| Code | Gender | Category (ICD-10, 2016) | Age (in years) | Cause of deficit |
|------|--------|-------------------------|----------------|------------------|
| 1 | M | 5 | 7 | Shaken Baby Syndrome |
| 2 | M | 5 | 8 | Cortical Blindness |
| 3 | M | 4 | 10 | Lowe Syndrome |
| 4 | M | 5 | 7 | Genetic Malformation |
| 5 | M | 4 | 7 | Immune Deficiency |
| 6 | M | 3 | 7 | Alstrom Syndrome |
| 7 | F | 3 | 9 | Unspecified |
| 8 | M | 3 | 12 | Unspecified |

capacities and familiarity with the visual conventions of drawing. However, prior contact with tactile imagery could tip the balance in favor of blind children [1].

## Method

### Participants

23 new children participated in Experiment 2: 8 early blind children and 15 sighted children. The early blind group was composed by 1 girl and 7 boys (M = 8.3 years, SD = 1.2, range 6 to 12 years) and the control group of sighted children by 6 girls and 9 boys (7.2 years, SD = 0.4, range 7 to 8 years).

The blind and sighted children tested in Experiment 2 attend the same schools and are in the same classes as those in Experiment 1. Parents or guardians also provided signed written informed consent forms for their children to participate in the experiment. The sighted children had normal or corrected to normal vision and were typically developing. The characteristics of early blind children are presented in Table 3. All are also included in one of the three categories of Blindness (Visual impairment categories 3, 4 or 5 in both eyes) of the International Statistical Classification of Diseases and Related Health Problems (ICD-10th revision, WHO 2016). They have no associated disorders, as reported by parents and teachers.

In order to determine the children's frequency of contact with tactile books, we asked the adult responsible for each child to complete a 5-point scale (0—*Never* to 4—*Everyday*) questionnaire. All early blind children had regular or moderate contact with tactile books (every day or several times a month). The mean level of use for blind children is 2.75 (SD = 0.88) and for sighted children is 0.67 (SD = 0.49).

### Stimuli

The set of textured pictures tested in Experiment 2 are 2D representations of the action objects included in Experiment 1 (the Swing, Bicycle, Toboggan, Trampoline, Merry-go-round, Roller Skate and Stairs). The set of pictures is presented in Fig 2. Stimuli were produced by the non-profit publishing house Les Doigts Qui Rêvent as they are currently found in tactile books for blind children. Each illustration board measured 20 cm x 20 cm. To control the relevance of 2D representational conventions in the set of stimuli used in this experiment, we assessed the identification of our illustrations by 10 blindfolded sighted adults (M = 25.2 years, SD = 3.7). Illustrations were presented randomly to participants. The same categorical information for each object was provided in the tests with the children. The results showed that on average, 80% (SD = 20%) of illustrations were correctly identified by blindfolded sighted adults. The least recognized illustration was the Merry-go-round (50% corrected identifications) and the most recognized were the Stairs and Toboggan (100% corrected identification for both).

### Familiarization shapes

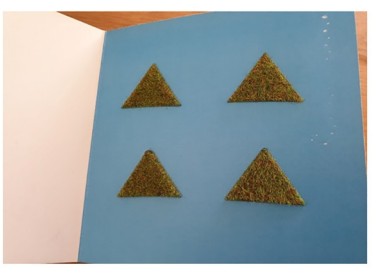 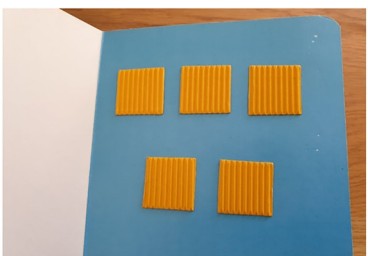

### Textured pictures of action objects

Swing

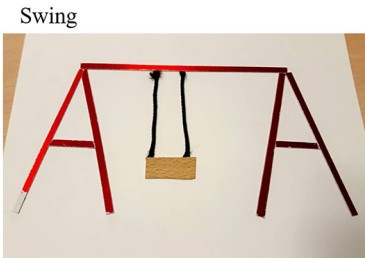

Bicycle

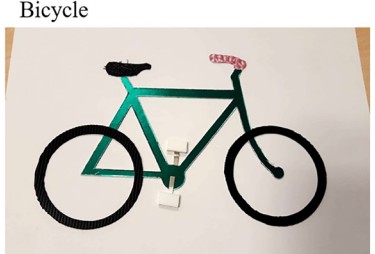

Toboggan

Trampoline

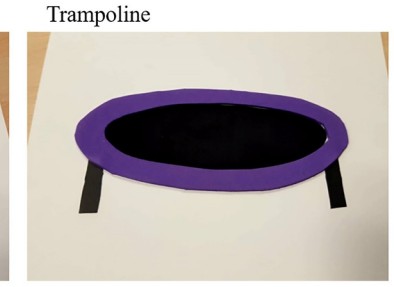

## Stimuli

Merry-go-round

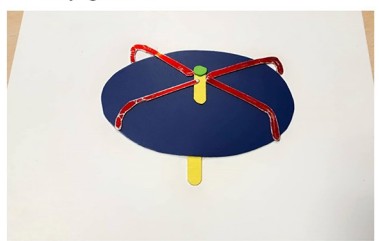

Roller skate

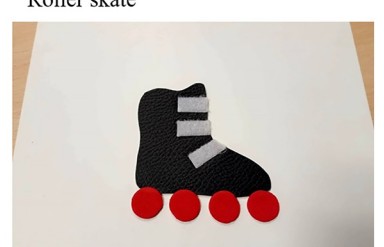

Stairs

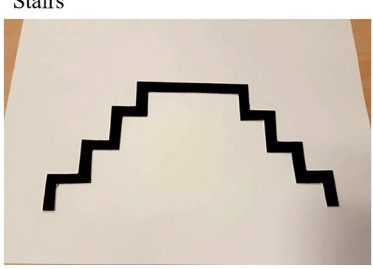

**Fig 2. Stimuli included in Experiment 2.** Fig 2A. Familiarisation shapes, Fig 2B. Textured picture of Swing, Bicycle, Toboggan and Trampoline, Fig 2C. Textured picture of Merry-go-round, Roller Skate and Stairs.

## Procedure

The procedure was the same as in Experiment 1. A familiarization phase with the perception of shapes by touch also took place before the test. The aim of this procedure was to minimize the disadvantages of sighted children compared to blind children due to the fact that they have less training in identification of shapes by touch. Thus, to all participants, sighted and blind children, we presented two pages of a tactile book containing geometric textured shapes (see Fig 2A Familiarization shapes). Page 1 contained 4 triangles depicted with a soft texture and page 2 contained 5 squares depicted with a rough texture. All blind and sighted children easily identified the 2 sets of geometric shapes by touch.

In the identification test, participants were told to freely explore each textured picture using both hands, and to try to identify the object represented. Sets of textured pictures were presented in a random order across participants. The experimenter gave the same categorical information about the objects as in Experiment 1.

Also as in Experiment 1, we carried out a free, one-minute identification phase followed by a guidance phase. In this guidance phase, the experimenter helped the child explore the completeness of the tactile elements in the picture. The experimenter motivated the child to define the shape of each tactile element touched (square, circle, oval, stick) without inducing any figurative interpretation of the elements. The experimenter touched all the tactile elements with the child once. If the child was still unable to identify the illustration even after the guidance phase, they were told to inform the experimenter when they wished to stop exploring.

The calculation of the scores and data processing were the same as in Experiment 1.

## Results

Mixed-design ANOVAs were performed on identification scores with action objects as within subjects factors (the Swing, Bicycle, Toboggan, Trampoline, Merry-go-round, Roller Skate and Stairs) and group as a between subjects factor (early blind vs sighted). Results showed an effect of group ($F(1,21) = 6.00$, $p = .023$). Surprisingly, blind children scored higher (score = 0.98, SD = 0.31) than sighted children (score 0.66, SD = 0.30) on this identification task. Blind children scored higher than sighted children for all action objects. Results also revealed a main effect of action object on identification scores ($F(6,126) = 8.33$, $p < .001$). Table 4 shows identification scores for each action object represented in textured pictures. It should be noted that identification performance varied considerably from one picture to another. The toboggan and Bicycle obtained the highest identification scores. A post-analysis using Bonferroni

**Table 4. Identification scores (max 2.00) and (SD) in sighted (N = 15) and early blind children (N = 8) for each action object represented with a textured picture.**

| Action Object | Sighted | | Early blind | | Total | |
|---|---|---|---|---|---|---|
| Swing | 0.60 | (0.63) | 1.13 | (0.99) | 0.78 | (0.80) |
| Bicycle | 1.00 | (0.66) | 1.63 | (0.74) | 1.22 | (0.74) |
| Toboggan | 1.27 | (0.80) | 1.75 | (0.46) | 1.43 | (0.73) |
| Trampoline | 0.67 | (0.82) | 0.25 | (0.46) | 0.52 | (0.73) |
| Merry-go-round | 0.20 | (0.41) | 0.88 | (0.99) | 0.43 | (0.73) |
| Roller Skate | 0.27 | (0.59) | 0.25 | (0.71) | 0.26 | (0.62) |
| Stairs | 0.60 | (0.83) | 1.00 | (1.07) | 0.74 | (0.92) |
| Total | 0.66 | (0.30) | 0.98 | (0.31) | | |

showed a significant difference only between objects obtaining the highest scores and those obtaining the lowest scores. More precisely, significant differences in scores were found between the Toboggan and the three other objects (the Trampoline, $p$ = .003; Merry-go-round, $p$ < .001 and Roller-Skate, $p$ < .001) and between the Bicycle, the second-best recognized object, and two other objects (the Merry-go-round, $p$ = .003 and Roller Skate, $p$ < .001). The interaction between action object and group was not significant (F(6, 126) = 1.59, p = .15).

## Discussion

Our hypothesis that sighted children would perform better than their early blind counterparts in this tactile picture identification task was not confirmed. Our reversed results observed in children (higher score in blind children) did not corroborate previous findings observed in adults [14,16]. The level of expertise with tactile images may explain this result. Indeed, all blind children who participated in our study had moderate to frequent contact with tactile illustrated books. Blind participants were recruited with the help of the non-profit publishing house Les Doigts Qui Revent, which put us in contact with educational center partners. Consequently, our sample is composed by children who are very stimulated and have a great deal of contact with tactile illustrated books. Several studies have shown that expertise with tactile images and the manual tactile exploratory process improves the identification and comprehension of tactile content [1,13,17,21,41]. Thus, the effect of practice can explain the better performances of blind children in our task, which globally remained very weak, as hypothesized.

The textured pictures with the highest identification scores in both groups were the Toboggan and Bicycle. Results obtained for the Toboggan corroborate the results of a previous study about the recognition of tactile drawings by blind children [19]. This previous study showed that tactile drawings containing trajectories of interaction with objects (the toboggan and stairs) were more easily understood by blind children than drawings that simply represented the formal properties of the objects themselves (a house or a cat). Interestingly, sensorimotor experiences also have a role to play here. We speculate that children were more able to identify the textured picture of the Toboggan because they explored the lines with their fingertips and this exploration mimicked the movement of the feet climbing up a ladder and the body sliding in an oblique line down the Toboggan. However, this assumption is not consistent with the Stairs in the present study. This can be explained by that fact that in previous studies, drawings of stairs showed only a rise in profile (like pictograms) whereas in our test the stairs were represented with the silhouette going up and down. A complete silhouette going up and down must also include the shape of a concrete object such as a house or a pyramid.

An explanation more related with cognitive categories and levels of typicality [42,43] may explain the high scores obtained for the Bicycle. It should be noted here that we provided the category of this object (to move) during the test. The Bicycle is a typical example of the "To move" object category. We also observed that for most of the participants (sighted or blind) the fact of having identified the two wheels of the bicycle was enough to arrive at the correct answer. Two wheels are a figurative attribute representative of the object Bicycle in drawing [44]. The Bicycle is a very common feature in children's educational books on the subject of means of transport, in illustrated tactile books for blind children as well.

## General discussion

The present study addresses the question of adapting tactile content for blind people. Considering the great and well-known difficulty encountered by blind subjects in identifying tactile pictures and drawings [1,9,14,16] we examined if a new embodied design of this content explored with ASFG [25] can improve the identification of objects. To assess this question, we

examined identification scores of 7 action objects using an ASFG procedure in Experiment 1 and the identification of the same action objects represented by textured pictures in Experiment (control) 2. To determine if visual status has an effect on identification abilities, performances of early blind and sighted children were compared in each experiment.

The results of Experiment 1 revealed that 3D miniatures of action objects explored with ASFG obtained high scores regardless of participants' visual status. According to our hypothesis, the data obtained suggests that the activation of real body experiences by two fingers that act as two legs in interaction with miniatures can help the tactile identification processes. We have already shown in a previous study [25] that blind and sighted people produced similar patterns of gestures to simulate an action with two fingers. The result of the present study reinforces these findings by showing that this symbolic "fingers-legs" relationship can also improve the identification of objects by touch. Our results provided evidence that a central idea of sensorimotor simulation from embodied cognition approaches [26–28] also has a great potential for designing materials for blind children. In addition to our results, it is interesting to note that during the task some blind children had a lot of fun adding sound effects to ASFG exploration (for instance, the *foushhh*—sound of sliding on a toboggan or the *boing boing* sound of jumping on a trampoline). The fact that blind children added complementary sensorial elements of real interaction with objects reinforces evidence that this new design invites them on a "journey" through real past sensorimotor experiences with objects.

In Experiment 2, an identification test of textured pictures was conducted with the same items in order to confirm the difficulty of this task and to reinforce evidence about the effect of the sensorimotor component in the identification of action objects. In each experiment the same categorical information was provided for each object. All objects obtained lower identification scores in Experiment 2 than in Experiment 1, except for the Bicycle. However, due to problems related to the tactile device itself (the same problems for the Roller Skate), it is difficult to know if the motor component was really involved in the identification of the Bicycle in Experiment 1. The typicality of the Bicycle in the category "To move" and the frequency with which this image appears in children's books contributed to increased identification scores of the Bicycle in Experiment 2.

One action object, the Toboggan, deviated from this tendency by obtaining high identification scores both in Experiment 1 and Experiment 2. Following results from a previous study [19] it is very interesting to note that the sensorimotor component of the Toboggan is also activated in the tactile picture of this object. Manual exploration of tactile patterns mimicked the movement of the feet climbing up a ladder and the body sliding in an oblique line down the toboggan. This result provided additional evidence that embodied experiences help the identification of objects by touch.

Finally, no differences on identification scores between sighted and blind children were found in Experiment 1. This finding is very important because it means that the ASFG procedure has a very inclusive potential to be relevant for a larger number of subjects, regardless of their visual skills. In contrast, differences between scores obtained by blind and sighted children in Experiment 1 corroborate studies that have shown that the ability to recognize 2D content by touch depends both on prior contact with tactile images [1,13,15,19,20] and training with bimanual exploratory processes [13,21].

A limitation to the generalization of the findings reported here, however, is the small sample size of early blind children included in the two experiments. Indeed, gathering samples of sufficient size is a common challenge for researchers working in the field of visual impairment [45,46]. The population of people with visual impairments is characterized by wide interpersonal variability due to the type of visual pathology and age of diagnosis. There is a small number of adults and children who are completely blind since birth or since the first few years of

life, and some have pathologies entailing additional handicaps. Thus, instead of trying to gather large samples of subjects at risk for having a large interpersonal variability, we found it more relevant to have two small samples with a larger exclusion criterion. This enables us to have corresponding inter-group profiles with regard to the age of diagnosis of complete blindness (early blind), no additional handicaps and educational level. Without intending to statistically compare results from the two experiments, the profile match between the two groups provides an overview of the effects of two techniques of illustration on early blind and sighted childrens' identification of the same objects. Although our simple size is comparable with other studies in the field assessing recognition of tactile content by blind children and adults [2,23,47], findings need to be taken into account carefully.

From a practical point of view, our study provides innovative solutions to teachers, parents and publishers looking for educational tools that are adapted to blind children's needs and at the same time shared between blind and sighted children.

The central question of our study is the contribution of embodied cognition and sensorimotor experiences in blind and sighted childrens' identification of illustrations. The advantage of 3D scenarios that engage ASFGs over traditional 3D miniatures (toys or miniature objects already commonly used in this field of intervention) is the reactivation of sensorimotor components by a gesture simulation (two fingers that act as two legs). As mentioned, this procedure was validated during the first phase of research in which a set of 18 ASFGs have been studied [25]. In the present study, we tested only 7 of these 18 following a feasibility criterion due to the design and nature of the study. Specifically, the ASFGs included should be reproducible in 3D miniature and also represented as 2D content. However, it is important to note that our previous study [25] showed that several other ASFGs are also very prototypical (same gestural pattern in the congenitally blind, late blind and sighted adults) such as "Kicking a ball", "Jumping on one leg" or "Walking backwards". These other ASFGs can be included in a book for blind children even if they have not been tested here.

Let us also add that the contribution of the embodied cognition that we have awakened via the ASFG procedure is not limited to the use of the ASFG. An example from our own study is the illustration of the toboggan as a textured picture that seems to be better recognized than the other illustrations due to the possible role of the sensorimotor experience reactivated when hands explore the slope of the toboggan. Other examples of illustrations involving sensorimotor or perceptual experiences other than visual ones can be found in books for blind children. For example, in the book entitled Wa-Wa (2012) about toilets published by the non-profit publishing house Les Doigts Qui Rêvent, children can flush the toilet thanks to an interactive system and a cord attached to the page. In A Magic Winter (2012) also published by Les Doigts Qui Rêvent, the child can walk on the snow, which is "illustrated" by a texture reproducing the sound and the sensation under the feet. The book "We're going on a bear hunt" (2015) designed with our scientific support [48] has a sliding mechanism allowing the reader to move a character to the other end of the page. In the very interactive book "The three little pigs" (2019) published by the non-profit publishing house Mes Mains en Or, the reader blows down pop-up houses with his breath.

Following direct feedback from child readers and their entourage who seem to appreciate this content more than the visual content in relief, some designers of tactile books are already choosing illustrations that allow for manipulation of objects, sounds and reactivation of haptic and sensorimotor experiences. The major contribution of our study is to provide scientific evidence of the role of these experiences in the identification of objects by touch, regardless of the visual status of the children. A feasibility study is already underway with our editorial partnership (non-profit publishing house Les Doigts Qui Rêvent) to evaluate the possibilities of including ASFGs in tactile devices such as children's books but also in mobility aids.

## Supporting information

**S1 Table. Appendix mean scores of contact frequency with action objects—Experiment 1.**
(DOCX)

**S2 Table. Data set Experiment 1.**
(XLSX)

**S3 Table. Data set Experiment 2.**
(XLSX)

## Acknowledgments

We would like to thank the children who took part in the study and all the educational and medico-social structures that welcomed us for the tests. We would also like to express our sincere thanks to the team of the non-profit publishing house Les Doigts Qui Rêvent, partner of this study, particularly Solène Négrerie for her contribution to the design of tactile prototypes. We would also like to thank Coralie Vouillon, Master's student of University of Geneva, for her participation in the collection of data and Lola Chennaz, Doctorate student of University of Geneva, for her participation in the final proofreading. English revision has been carried out by Susan Campbell.

## Author Contributions

**Conceptualization:** Dannyelle Valente, Edouard Gentaz.

**Data curation:** Dannyelle Valente.

**Formal analysis:** Dannyelle Valente, Amaya Palama, Edouard Gentaz.

**Funding acquisition:** Edouard Gentaz.

**Investigation:** Dannyelle Valente.

**Methodology:** Dannyelle Valente, Amaya Palama, Edouard Gentaz.

**Project administration:** Edouard Gentaz.

**Writing – original draft:** Dannyelle Valente, Amaya Palama, Edouard Gentaz.

**Writing – review & editing:** Dannyelle Valente, Amaya Palama, Edouard Gentaz.

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
