## [Decision Letter · Decision Letter 0]

17 Nov 2020

PONE-D-20-13459

Exploring 3D miniatures with action simulations by finger gestures: study of a new embodied design for blind and sighted children

PLOS ONE

Dear Dr. Valente,

Thank you for submitting your manuscript to PLOS ONE - please accept my apologies for the time it has taken us to get a decision to you.

Your manuscript has been evaluated by two external reviewers, whose reports are appended to this letter. The reports are broadly positive, but also raise some important concerns regarding the framing of the conclusions and study limitations.

After careful consideration of the reports, we feel that your manuscript has merit but does not fully meet PLOS ONE’s publication criteria as it currently stands. Therefore, we invite you to submit a revised version of the manuscript that addresses the points raised during the review process.

We look forward to receiving your revised manuscript.

Kind regards,

Dr Joseph Donlan

Senior Editor

PLOS ONE

Journal Requirements:

2. Please state in your methods section whether you obtained consent from parents or guardians of the minors (those aged <18) included in the study or whether the research ethics committee or IRB approved the lack of parent or guardian consent.

Reviewers' comments:

Reviewer's Responses to Questions

**Comments to the Author**

1. Is the manuscript technically sound, and do the data support the conclusions?

Reviewer #1: Partly

Reviewer #2: Yes

2. Has the statistical analysis been performed appropriately and rigorously? 

Reviewer #1: Yes

Reviewer #2: Yes

3. Have the authors made all data underlying the findings in their manuscript fully available?

Reviewer #1: Yes

Reviewer #2: Yes

4. Is the manuscript presented in an intelligible fashion and written in standard English?

Reviewer #1: Yes

Reviewer #2: Yes

5. Review Comments to the Author

Reviewer #1: This paper explores how action simulations by finger gestures (e.g., moving two fingers on a swing) can improve the identification of objects by children with visual impairments. This work has the motivation to improve the development of books for children with visual impairments as well as other materials where object identification is relevant.

The paper builds upon research by the same authors on identification of objects by blind adults. The background is clear and comprehensive, and the work is very well motivated.

The paper presents two experiments: one where the proposed procedure is compared in object identification tasks between blind and sighted children, and a second one where texture pictures (the de facto method for conveying tactile information to blind children) are compared in identification tasks by a set of different sighted and blind children.

There is one overarching question related to the motivation presented by the authors that relates to the number of actions that can be presented with ASFG. It would be relevant to understand how this procedure can be applied beyond the limited set of actions presented. I am sure there are others, but question if they are representative in the large number of objects available in children's books. This is not a deal breaker for the specific contribution that is offered in this manuscript, but something that this reviewer finds relevant to see discussed given the context that is offered in the introduction.

The major concern I have with the paper is the low sample size in each of the experiments, particularly considering visually impaired children (n=8 in each experiment). Adding to that, the experiments were also performed with different children. Although the paper does not try to explictly and statistically compare results between the two "experiments", it implies that recognition are better with ASFG than with Textures. Given the variability of children with visual impairments, and that not attempt to match the experimental and control groups was made, it is impossible to say something about this comparison. The results within each experiment are interesting, but given the limited sample sizes, they need to be taken into account carefully.

I do find this work relevant and fidn the experiments to be interesting preliminary investigations. However, I would like to see the aforementioned points considered, and limitations clearly presented, toning down conclusions when needed.

Minor issues:

Please avoid using terms like "the blind" as they tend to define a person by their disability. Use instead terms like "blind people" or "people who are blind" (http://interactions.acm.org/archive/view/november-december-2015/writing-about-accessibility)

83: Close parenthesis

108: This idea is using 3D

Reviewer #2: The paper presented an original study on 3D sensorimotor miniature explorations for embodied understanding of complex shapes and textures for visually impaired and blindfolded children. The topic is original, the state of art is quite extensive, even if I suggest to authors to add recent studies on sensory integration in typically and visually impaired children (e.g. Cappagli et al., 2019, Gori et al. 2010). The methodology and data analysis are well-explained and the paper is overall well written. For all these reason, I recommend to accept this contribution for publication.

6. PLOS authors have the option to publish the peer review history of their article (what does this mean?). If published, this will include your full peer review and any attached files.

Reviewer #1: No

Reviewer #2: **Yes: **Erica Volta

---

## [Author Response · Author response to Decision Letter 0]

4 Dec 2020

Response to reviewers for “Exploring 3D miniatures with action simulations by finger gestures: study of a new embodied design for blind and sighted children” (PONE-D-20-13459)

We would like to thank the editor and the reviewers for their constructive comments. We addressed all the points raised by the reviewer and editor and explained how below. In the manuscript, all changes are marked in blue. 

Journal Requirements:

Answer : 

In title page, country of authors are included. We proceeded to a verification of others items and style requirements are respected. 

2. Please state in your methods section whether you obtained consent from parents or guardians of the minors (those aged <18) included in the study or whether the research ethics committee or IRB approved the lack of parent or guardian consent.

Answer : 

The statement “Parents or guardians provided signed written informed consent forms for their children to participate in the experiment “ was included for Experiment 1 (page 10, line 205) and for Experiment 2 (page 18, line 369)

Reviewer #1: 

Reviewer #1 : There is one overarching question related to the motivation presented by the authors that relates to the number of actions that can be presented with ASFG. It would be relevant to understand how this procedure can be applied beyond the limited set of actions presented. I am sure there are others, but question if they are representative in the large number of objects available in children's books. This is not a deal breaker for the specific contribution that is offered in this manuscript, but something that this reviewer finds relevant to see discussed given the context that is offered in the introduction.

Answer:

Thank you for raising this very interesting point. Indeed, although we explain in the Method section (page 10, line 211) the criteria on which we based our selection of the 7 ASFGs in this study, it is very important to show how these conclusions drawn from a limited set of actions can be extended to other illustrations included in books for blind children. We therefore further develop this point in General Discussion: 

Page 24, line 543: The central question of our study is the contribution of embodied cognition and sensorimotor experiences in blind and sighted childrens’ identification of illustrations. The advantage of 3D scenarios that engage ASFGs over traditional 3D miniatures (toys or miniature objects already commonly used in this field of intervention) is the reactivation of sensorimotor components by a gesture simulation (two fingers that act as two legs). As mentioned, this procedure was validated during the first phase of research in which a set of 18 ASFGs have been studied [25]. In the present study, we tested only 7 of these 18 following a feasibility criterion due to the design and nature of the study. Specifically, the ASFGs included should be reproducible in 3D miniature and also represented as 2D content. However, it is important to note that our previous study [25] showed that several other ASFGs are also very prototypical (same gestural pattern in the congenitally blind, late blind and sighted adults) such as "Kicking a ball", "Jumping on one leg" or "Walking backwards". These other ASFGs can be included in a book for blind children even if they have not been tested here. 

Let us also add that the contribution of the embodied cognition that we have awakened via the ASFG procedure is not limited to the use of the ASFG. An example from our own study is the illustration of the toboggan as a textured picture that seems to be better recognized than the other illustrations due to the possible role of the sensorimotor experience reactivated when hands explore the slope of the toboggan. Other examples of illustrations involving sensorimotor or perceptual experiences other than visual ones can be found in books for blind children. For example, in the book entitled Wa-Wa (2012) about toilets published by the non-profit publishing house Les Doigts Qui Rêvent, children can flush the toilet thanks to an interactive system and a cord attached to the page. In A Magic Winter (2012) also published by Les Doigts Qui Rêvent, the child can walk on the snow, which is “illustrated” by a texture reproducing the sound and the sensation under the feet. The book “We’re going on a bear hunt” (2015) designed with our scientific support [48] has a sliding mechanism allowing the reader to move a character to the other end of the page. In the very interactive book “The three little pigs” (2019) published by the non-profit publishing house Mes Mains en Or, the reader blows down pop-up houses with his breath. 

Following direct feedback from child readers and their entourage who seem to appreciate this content more than the visual content in relief, some designers of tactile books are already choosing illustrations that allow for manipulation of objects, sounds and reactivation of haptic and sensorimotor experiences. The major contribution of our study is to provide scientific evidence of the role of these experiences in the identification of objects by touch, regardless of the visual status of the children. A feasibility study is already underway with our editorial partnership (non-profit publishing house Les Doigts Qui Rêvent) to evaluate the possibilities of including ASFGs in tactile devices such as children's books but also in mobility aids. 

Reviewer #1: The major concern I have with the paper is the low sample size in each of the experiments, particularly considering visually impaired children (n=8 in each experiment). Adding to that, the experiments were also performed with different children. Although the paper does not try to explicitly and statistically compare results between the two "experiments", it implies that recognition are better with ASFG than with Textures. Given the variability of children with visual impairments, and that not attempt to match the experimental and control groups was made, it is impossible to say something about this comparison. The results within each experiment are interesting, but given the limited sample sizes, they need to be taken into account carefully.

Answer: 

Thank you for this comment. Despite our sample size being comparable with other studies in the field (Lederman et al, 1990; Bara, 2014; Heller, 1996), we agree that the sample size of early blind children participated in two experiments is a limitation to the generalization of the findings. Hence, we added a limitation paragraph at the discussion to point this issue: 

Page 25, line 524: A limitation to the generalization of the findings reported here, however, is the small sample size of early blind children included in the two experiments. Indeed, gathering samples of sufficient size is a common challenge for researchers working in the field of visual impairment [45, 46]. The population of people with visual impairments is characterized by wide interpersonal variability due to the type of visual pathology and age of diagnosis. There is a small number of adults and children who are completely blind since birth or since the first few years of life, and some have pathologies entailing additional handicaps. Thus, instead of trying to gather large samples of subjects at risk for having a large interpersonal variability, we found it more relevant to have two small samples with a larger exclusion criterion. This enables us to have corresponding inter-group profiles with regard to the age of diagnosis of complete blindness (early blind), no additional handicaps and educational level. Without intending to statistically compare results from the two experiments, the profile match between the two groups provides an overview of the effects of two techniques of illustration on early blind and sighted childrens’ identification of the same objects. Although our simple size is comparable with other studies in the field assessing recognition of tactile content by blind children and adults [2, 23, 47], findings need to be taken into account carefully. 

Reviewer #1/Minor issues : Please avoid using terms like "the blind" as they tend to define a person by their disability. Use instead terms like "blind people" or "people who are blind" (http://interactions.acm.org/archive/view/november-december-2015/writing-about-accessibility) : 

Line 37: “The blind” changed for “blind people” 

Line 156: “the sighted and the blind simulations” changed for “the sighted and blind adults’ simulations”

Line 240: the blind changed for “blind people”

83: Close parenthesis: 

Line 83, Corrected

108: This idea is using 3D: 

Line 108: Corrected, “the idea is to use” 

Reviewer #2: I suggest to authors to add recent studies on sensory integration in typically and visually impaired children (e.g. Cappagli et al., 2019, Gori et al. 2010). 

 Answer: 

Thank you very much for this suggestion. Indeed, our study is in dialogue with the proposed references, which show the importance of multisensory calibration in a context of blindness. We have included these studies in the introduction:

Page 6, line 129 : Furthermore, studies on sensory integration [37, 38] have shown that lack of vision can alter the intermodal calibration necessary for spatial discrimination in blind children [for a review see 39]. Thus, compensation via other intact senses associating for example auditory and motor information can help blind children to process spatial information [38]. Gappagli and colleagues have tested a technological device (ABBI - Audio Bracelet for Blind Interaction), which emits a sound when a movement occurs, with 3-7 year-old children with congenital blindness or low vision. After 3 months of training, authors found that the spatial performance of congenitally blind but not low vision children is improved. Results indicated that in the absence of sight, the association of sensorimotor data with other intact senses is of great importance.

37. Gori M, Sandini G, Martinoli C, Burr D. Impairment of auditory spatial localization in congenitally blind human subjects. Brain. 2014;137(1):288-93. doi: doi.org/10.1093/brain/awt311.

38. Cappagli G, Finocchietti S, Baud-Bovy G, Cocchi E, Gori M. Multisensory rehabilitation training improves spatial perception in totally but not partially visually deprived children. Frontiers in Integrative Neuroscience. 2017;11. doi: doi.org/10.3389/fnint.2017.00029.

39. Valente D, Bara F, Afonso Jaco A, Baltenneck N, Gentaz E. La perception tactile des propriétés spatiales des objets chez les personnes aveugles. Enfance. in press.

---

## [Editor Report · Decision Letter 1]

2 Jan 2021

Exploring 3D miniatures with action simulations by finger gestures: study of a new embodied design for blind and sighted children

PONE-D-20-13459R1

Dear Dr. Valente,

We’re pleased to inform you that your manuscript has been judged scientifically suitable for publication and will be formally accepted for publication once it meets all outstanding technical requirements.

Kind regards,

Tiago Guerreiro, Ph.D.

Guest Editor

PLOS ONE

Additional Editor Comments:

For full disclosure, I participated as a reviewer for the initial evaluation of this manuscript, as Reviewer 1, and asked for clarifications and changes. I appreciate the changes performed in the manuscript; I find the paper to be methodologically sound and clear.

---

## [Editor Report · Acceptance letter]

7 Jan 2021

PONE-D-20-13459R1 

Exploring 3D miniatures with action simulations by finger gestures: study of a new embodied design for blind and sighted children 

Dear Dr. Valente:

I'm pleased to inform you that your manuscript has been deemed suitable for publication in PLOS ONE. Congratulations! Your manuscript is now with our production department. 

Kind regards, 

on behalf of

Dr. Tiago Guerreiro 

Guest Editor

PLOS ONE